# Calcium transients trigger switch-like discharge of prostaglandin E$_2$ in an extracellular signal-regulated kinase-dependent manner

**Tetsuya Watabe[1,2], Shinya Yamahira[1], Kanako Takakura[1], Dean Thumkeo[3], Shuh Narumiya[3], Michiyuki Matsuda[1,2,4], Kenta Terai[2]***

[1]Research Center for Dynamic Living Systems, Graduate School of Biostudies, Kyoto University, Kyoto, Japan; [2]Department of Pathology and Biology of Diseases, Graduate School of Medicine, Kyoto University, Kyoto, Japan; [3]Department of Drug Discovery Medicine, Graduate School of Medicine, Kyoto University, Kyoto, Japan; [4]Institute for Integrated Cell-Material Sciences, Kyoto University, Kyoto, Japan

**\*For correspondence:** terai.kenta.5m@tokushima-u.ac.jp

**Abstract** Prostaglandin E$_2$ (PGE$_2$) is a key player in a plethora of physiological and pathological events. Nevertheless, little is known about the dynamics of PGE$_2$ secretion from a single cell and its effect on the neighboring cells. Here, by observing confluent Madin–Darby canine kidney (MDCK) epithelial cells expressing fluorescent biosensors, we demonstrate that calcium transients in a single cell cause PGE$_2$-mediated radial spread of PKA activation (RSPA) in neighboring cells. By in vivo imaging, RSPA was also observed in the basal layer of the mouse epidermis. Experiments with an optogenetic tool revealed a switch-like PGE$_2$ discharge in response to the increasing cytoplasmic Ca$^{2+}$ concentrations. The cell density of MDCK cells correlated with the frequencies of calcium transients and the following RSPA. The extracellular signal-regulated kinase (ERK) activation also enhanced the frequency of RSPA in MDCK and in vivo. Thus, the PGE$_2$ discharge is regulated temporally by calcium transients and ERK activity.

## eLife assessment

This **important** study reports on the dynamics of PKA investigated at the single-cell level in vitro and in epithelia in vivo. Using different fluorescent biosensors and optogenetic actuators, the authors dissect the signaling pathway responsible for PKA waves, finding that PKA activation is a consequence of PGE$_2$ release, which in turn is triggered by calcium pulses, requiring high ERK activity. The evidence supporting the claims is **solid**. At this stage, the work is still partly descriptive in nature, and additional measurements would increase the strength of mechanistic insights and physiological relevance.

## Introduction

Prostaglandin E$_2$ (PGE$_2$) is an eicosanoid lipid mediator that regulates a plethora of homeostatic functions, including vascular permeability, immune response, and mucosal integrity (**Narumiya, 2007**). The metabolic pathway of PGE$_2$ production, which is a major branch of the arachidonic acid cascade, was extensively studied in the mid to late 20th century. This cascade starts from the activation of cytosolic phospholipase A2 (cPLA2) (**Park et al., 2006**). Increased intracellular calcium induces the translocation and activation of cPLA2 from the cytosol to the Golgi, endoplasmic reticulum, and perinuclear

membrane, where cPLA2 cleaves arachidonic acid out of the membrane phospholipids (*Clark et al., 1995*; *Hirabayashi et al., 1999*; *Evans et al., 2001*). The arachidonic acid is then presented to cyclo-oxygenases, COX1 and COX2, to yield $PGH_2$, which is further converted to $PGE_2$ by prostaglandin E synthases (*Smith and Langenbach, 2001*). $PGE_2$ synthesized de novo is secreted to the extracellular space either by passive diffusion or by active transport by multidrug resistance protein 4 (MRP4) (*Reid et al., 2003*). The secreted $PGE_2$ exerts its actions by acting on four G protein-coupled receptors (GPCRs), EP1 to EP4, expressed in neighboring cells (*Narumiya, 2007*; *Regan, 2003*).

Although it is largely believed that the $PGE_2$ action is primarily regulated by the expression and activation of COX (*Kalinski, 2012*), cPLA2 appears to play a more important role in the short-term regulation of $PGE_2$ production and secretion (*Leslie, 2015*). It was shown that the secretion of arachidonic acid is induced within a few minutes after the calcium-dependent translocation of cPLA2 to endo-membranes (*Hirabayashi et al., 1999*; *Evans et al., 2001*). Moreover, ERK and p38 MAP kinases also contribute to the activation of cPLA2 (*Lin et al., 1993*), which may be calcium-independent (*Gijón et al., 2000*). Importantly, these earlier biochemical studies did not elucidate the dynamics of the production and secretion of $PGE_2$ at the single-cell level, leaving many questions unanswered. For example, do all cells contribute to the production of $PGE_2$? Does each cell keep secreting $PGE_2$ upon stimulation?

Genetically encoded biosensors based on the principle of Förster resonance energy transfer (FRET) allow us to visualize the dynamics of intracellular signaling molecules at the single-cell resolution (*Miyawaki and Niino, 2015*; *Greenwald et al., 2018*). The development of transgenic mice expressing FRET biosensors has opened a window to the visualization of signaling molecule activity in live tissues (*Terai et al., 2019*). Furthermore, by using the activation of protein kinase A (PKA) and ERK MAP kinase as surrogate markers, we can also visualize the intercellular communications mediated by Gs-coupled receptors and tyrosine kinase receptors in live tissues (*Konishi et al., 2021*; *Hino et al., 2020*). In this study, we show that $PGE_2$ discharged from a single cell causes radial spread of PKA activation (RSPA) in neighboring cells in tissue culture and the mouse epidermis. By combining a chemical biology approach, optogenetic stimulation, and a simulation model, we quantitatively analyzed the $PGE_2$ secretion and found that the $PGE_2$ discharge is regulated temporally by calcium transients and quantitatively by growth factor signaling and cell density.

## Results

### $PGE_2$ mediates RSPA

In an attempt to understand intercellular communication under physiological conditions, we observed the PKA activity of Madin–Darby canine kidney (MDCK) epithelial cells by using the FRET biosensor Booster-PKA (*Zhang et al., 2001*; *Watabe et al., 2020*; *Figure 1A*). We noticed that PKA activation propagates from a single cell to neighboring cells under a confluent condition. We named this phenomenon radial spread of PKA activation (RSPA) and pursued underlying mechanisms (*Figure 1B*, *Figure 1—video 1*). In a typical example, approximately 100 neighboring cells located within a 100 μm distance exhibit firework-like spread of PKA activation, which decays within several minutes. To characterize this phenomenon without preoccupations, we developed a program to identify and characterize RSPA under various conditions (*Figure 1C*). The frequency, but not the radius, of RSPA depended on the cell density; that is, RSPA was observed only when cells were maintained at more than $6 \times 10^4$ cells/cm$^2$ (*Figure 1D and E*). The probability of RSPA in each cell was also increased in a cell density-dependent manner (*Figure 1—figure supplement 1A*). To examine whether the PKA activation correlates with increased intracellular cAMP concentration, we employed another FRET biosensor for cAMP, hyBRET-Epac, and performed a similar experiment (*Watabe et al., 2020*; *Ponsioen et al., 2004*; *Figure 1—figure supplement 1B and C*). Although the increment of the FRET ratio was not so remarkable as that of Booster-PKA (*Figure 1—figure supplement 1D*), we found that the pattern of cAMP concentration change is very similar to the activity change of PKA, indicating that a Gs protein-coupled receptor (GsPCR) mediates RSPA (*Figure 1—figure supplement 1E*). This discrepancy between hyBRET-Epac and Booster-PKA may be partially explained by the difference in the dynamic ranges for cAMP signaling in each FRET biosensor (*Watabe et al., 2020*). Previous transcriptome analysis of MDCK cells showed that ATP receptor and $PGE_2$ receptor EP2 are the most abundant GsPCRs (*Shukla et al., 2015*). Thus, we examined the contribution of the ATP receptor and

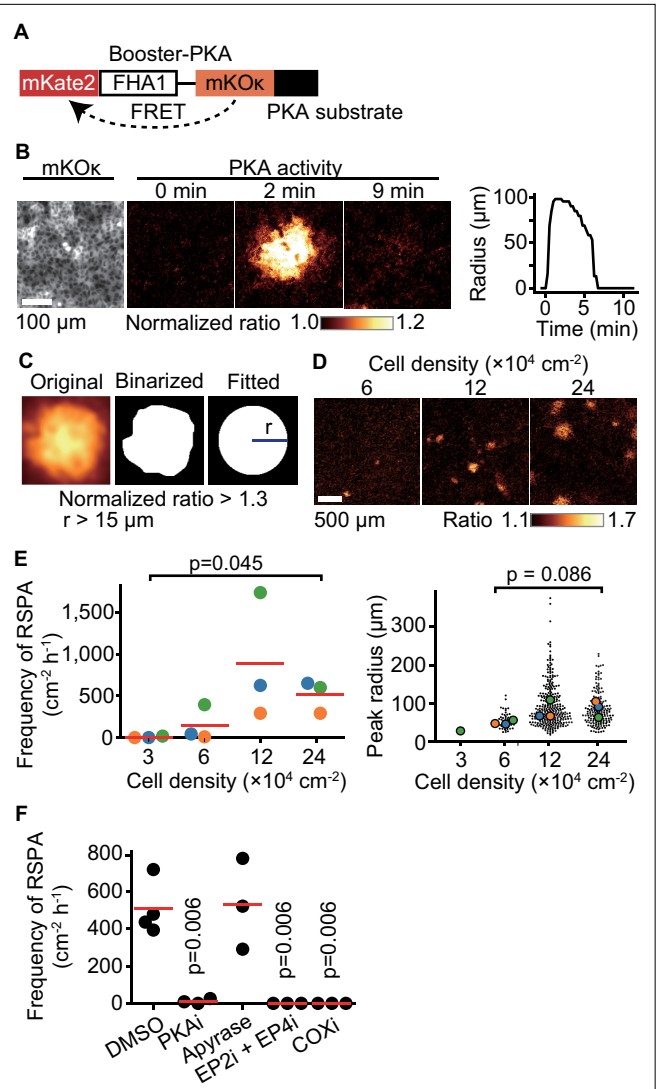

**Figure 1.** Radial spread of PKA activation (RSPA) in Madin–Darby canine kidney (MDCK) cells. (**A**) A scheme of Booster-PKA, a PKA sensor. (**B**) Booster-PKA-expressing MDCK cells in a confluent condition were observed every 15 s under a fluorescence microscope (**Figure 1—video 1**). The image of mKO $\kappa$ represents the cell density, which is seeded at $2.4 \times 10^5$ cells cm$^{-2}$. mKate2 and mKO $\kappa$ images were acquired to generate mKate2/mKO $\kappa$ ratio images representing PKA activity in pseudocolor. The time 0 is set the just before initiation of PKA activation. The radius of RSPA as determined in (**C**) was plotted as a function of time. (**C**) Procedure to call RSPA positive. The original ratio images were binarized with the threshold value 1.3 of mKate2/mKO $\kappa$ ratio. The fitted radius of RSPA, r, was defined as the radius of a circle with the same area. When r is >15 μm, it is counted as RSPA. The detailed procedure is provided in the 'Materials and methods' section. (**D, E**) MDCK cells expressing Booster-PKA were seeded at the indicated density and analyzed. Representative images in indicated cell densities are shown in pseudocolor (**D**). Each color in panel (**E**) represents an individual experiment. Red lines indicate average values. (**F**) MDCK cells expressing Booster-PKA in the presence of the inhibitors were imaged and analyzed for the RSPA frequency. Reagents are as follows: DMSO, 0.1% v/v DMSO; PKAi, 20 μM H89; Apyrase, 10 unit mL$^{-1}$; EP2i, 10 μM PF-04418948; EP4i, 1 μM ONO-AE3-208; COXi, 10 μM indomethacin. The frequency of RSPA was analyzed 20–80 min after the treatment. Each dot represents an individual experiment. Red lines indicate their average value. p-Values were calculated between the labeled sample and the DMSO-treated sample.

The online version of this article includes the following video and figure supplement(s) for figure 1:

**Figure supplement 1.** The probability of radial spread of PKA activation (RSPA) in each cell.

**Figure 1—video 1.** Radial spread of PKA activation (RSPA) in Madin–Darby canine kidney (MDCK) cells.

https://elifesciences.org/articles/86727/figures#fig1video1

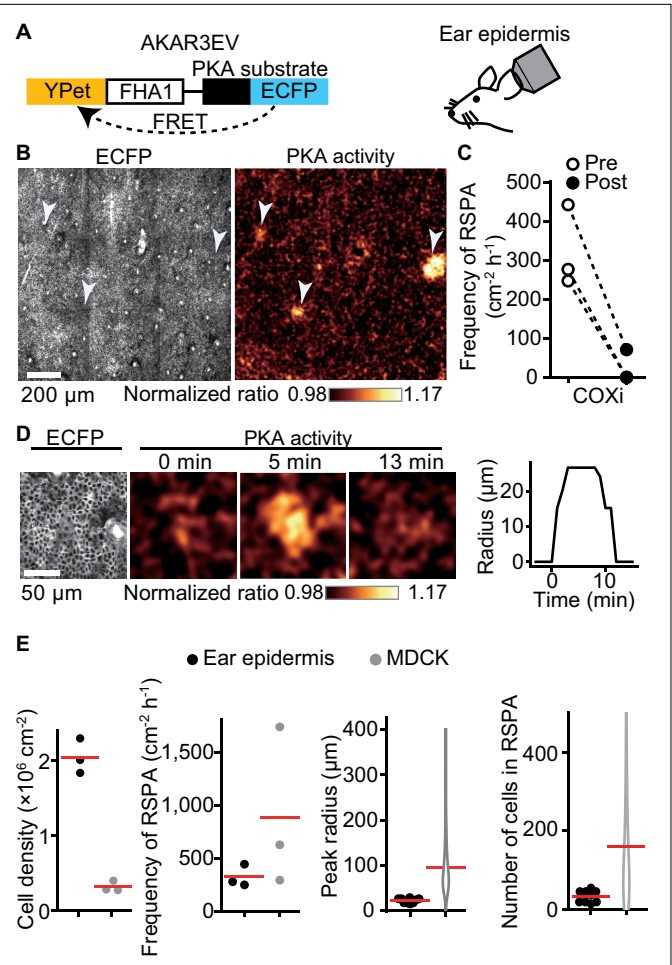

**Figure 2.** Radial spread of PKA activation (RSPA) in the basal layer of mouse auricular epidermis. (**A**) A scheme of AKAR3EV, a PKA sensor. (**B**) Transgenic mice expressing AKAR3EV were observed under a two-photon excitation microscope. Shown are an ECFP image and a YPet/ECFP ratio image representing the cell density and PKA activity in pseudocolor, respectively. (**C**) Three mice were administrated a COXi, 30 mg/kg flurbiprofen intraperitoneally. The frequency of RSPA in pretreatment was quantitated more than 40 min before the injection. Similarly, the frequency of a post-COX inhibitor treatment was analyzed 15–58, 15–97, and 15–63 min. Each dotted line represents an individual mouse experiment. (**D**) Magnified views of an RSPA in vivo. Shown are an ECFP image and a YPet/ECFP ratio image representing the cell density and PKA activity in pseudocolor, respectively. The radius was determined as in *Figure 1C*, with the detection limit of 6.4 μm. (**E**) The properties of RSPA are compared between in vivo and in vitro. Data from three independent experiments are shown for each condition. The data of MDCK is from *Figure 1E*, seeded at the 1.2 × 10⁵ cells cm⁻². Cell density and frequency of RSPA data are values per experiment. Peak radius data are pooled from three independent experiments for each condition. Red lines indicate average values.

The online version of this article includes the following video for figure 2:

**Figure 2—video 1.** Radial spread of PKA activation (RSPA) in living mice.
https://elifesciences.org/articles/86727/figures#fig2video1

---

PGE$_2$ receptors EP2 and EP4 by using specific inhibitors (*Figure 1F*). Inhibitors against EP2, EP4, and COX, but not the ATPase apyrase, abolished RSPA, indicating that PGE$_2$ mediates RSPA.

## RSPA is also observed in the epidermis of the PKAchu mice

To clarify the physiological relevance of RSPA, we used PKAchu mice, which are transgenic mice expressing a FRET biosensor for PKA, AKAR3EV (*Kamioka et al., 2012*; *Sato et al., 2020*; *Figure 2A*). We previously observed that ERK MAP kinase activation is propagated radially among the basal layer cells of the mouse epidermis (*Hiratsuka et al., 2015*), but we failed to observe a similar propagation

of PKA activation. We reasoned that this failure was due to the low frequency and short duration of this phenomenon. When we observed a region of over 1 mm square at 1 min intervals, we successfully observed RSPA in the basal layer of the mouse auricular epidermis (*Figure 2B*, *Figure 2—video 1*). Upon i.p. injection of a COX inhibitor, RSPA almost completely disappeared within 10 min (*Figure 2C*), indicating that RSPA in the epidermis is mediated by prostaglandins, presumably PGE$_2$. The cell density in the basal layer is approximately $2 \times 10^6$ cells cm$^{-2}$, which is markedly higher than that in MDCK cells (*Figure 2D and E*). It is not clear whether this may be related to the lower frequency (~300 cm$^{-2}$ hr$^{-1}$) and smaller radius of RSPA in the basal layer cells compared to MDCK cells (*Figure 2E*).

## RSPA is triggered by calcium transients

What causes RSPA? In agreement with the principal role of calcium in cPLA2 activation, dual imaging of calcium and PKA showed that an intracellular calcium transient precedes RSPA (*Figure 3A*). As anticipated, the frequency of RSPA was suppressed by the calcium chelator BAPTA-AM (*Figure 3B*). Note that not all of the calcium transients induced RSPA (*Figure 3C*, arrowheads). Approximately only one-tenth of calcium transients evoke RSPA (compare *Figures 1E and 3D*). Moreover, the frequency of calcium transients was also cell density-dependent (*Figure 3D*). To further pursue the relationship between calcium transient and RSPA, we employed the Gq-DREADD system (*Armbruster et al., 2007*). The Gq-DREADD-expressing producer cells were plated with the reporter cells expressing Booster-PKA at a 1:1200 ratio (*Figure 3E*). Upon activation of the Gq protein-coupled receptor by the DREADD ligand CNO, we observed RSPA with almost the same size and time course as observed under the non-stimulated condition (*Figure 3F*). Among cells with the calcium transient, 76% exhibited RSPA, significantly higher than that in the unstimulated state. Because the average calcium signal intensity was higher in RSPA (+) cells than in RSPA (-) cells (*Figure 3G*), the peak value of calcium transients appears to be important for the following induction of RSPA.

## RSPA is a switch-like response to cytoplasmic calcium concentration

To further explore the relationship between the peak value of the calcium transient and RSPA, we employed OptoSTIM1, an optogenetic tool to activate the calcium influx (*Kyung et al., 2015*; *Figure 4A*). As anticipated, a blue light flash caused a calcium transient, followed by RSPA (*Figure 4B*). In a preliminary experiment, a 30 min interval was sufficient to restore the calcium response; therefore, we repeated the blue light flash every 30 min to 1 hr with increasing light intensity. As the light intensity was increased, the amplitude of calcium transients represented by the R-GECO signal also increased linearly (*Figure 4C*). In stark contrast, RSPA occurred in an all-or-nothing manner. We repeated this experiment for 13 cells to find any correlations between the calcium concentration and the size of RSPA (*Figure 4D*). The R-GECO signal intensity ratio (F/F0) that evoked RSPA ranged from 1.5 to 2.1 among the different cells (*Figure 4D*, left), but the size of RSAP did not show a clear correlation with the R-GECO signal intensity ratio (*Figure 4D*, right). This observation indicates that there is a threshold of the cytoplasmic calcium concentration for the triggered PGE$_2$ secretion.

## High cell density increases the sensitivity to PGE$_2$

We next explored the mechanism that determines the size of RSPA. First, taking advantage of the reproducibility of DREADD system, we examined the involvement of EP2/EP4, cPLA2, and COX1/2 in the calcium-induced RSPA by CRISPR/Cas9-mediated gene knockout (*Figure 5A*). As anticipated, we did not observe any RSPA by using the reporter MDCK cells deficient from EP2 and EP4. Knockout of COX2 and cPLA2, but not COX1, in the producer cells almost completely abolished CNO-induced RSPA. These results support the idea that RSPA is mediated by PGE$_2$ via the Ca$^{2+}$-cPLA2-COX2-EP2/4-cAMP-PKA pathway (*Figure 5B*).

Next, because the size of RSPA depends on the cell density (*Figure 1E*), we reasoned that the sensitivity of MDCK cells to PGE$_2$ may also be regulated by cell density. To avoid the effect of PGE$_2$ produced by the cells, the COX1/2-deficient MDCK cells were challenged by the bath application of PGE$_2$. We found an approximately tenfold difference in the EC50 between the high and low cell densities (*Figure 5C*), suggesting that increased sensitivity to PGE$_2$ underlies the increased RSPA size under the confluent condition. Transcriptome analysis showed a twofold increase in guanine nucleotide-binding protein G(s) subunit alpha (GNAS) (*Figure 5D*), but it is not clear whether this difference is sufficient to explain the difference in RSPA frequency. We did not observe any cell density

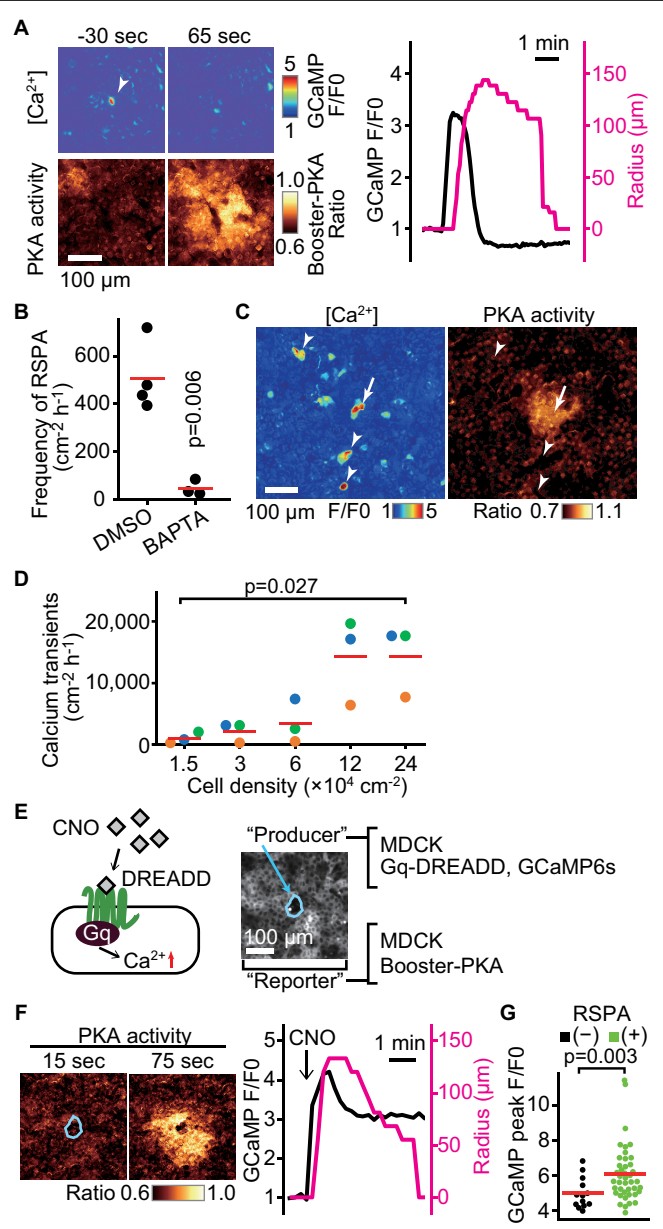

**Figure 3.** Radial spread of PKA activation (RSPA) induction by calcium transients. (**A**) Madin–Darby canine kidney (MDCK) cells expressing GCaMP6s and Booster-PKA were observed for calcium transients and RSPA. Calcium transients are represented by the fluorescence of GCaMP normalized to the basal level (F/F0). RSPA is analyzed as in *Figure 1C*. (**B**) The frequency of RSPA was analyzed 20–80 min after 30 μM BAPTA-AM treatment. Data from three or four independent experiments. The DMSO control data set is from *Figure 1F*. (**C**) Maximum projection images of the ratio over 10 min. An arrow or arrowheads represent calcium transients with or without RSPA, respectively. Shown is a part of *Figure 3—video 1*. (**D**) MDCK cells expressing GCaMP6s were seeded at the indicated density. Calcium transients showing F/F0 to be >3 were counted in an indicated cell density. (**E**) Schematic representation of RSPA induction using Gq-DREADD. MDCK cells expressing Gq-DREADD served as producer cells, while MDCK cells expressing Booster-PKA were employed as reporter cells. (**F, G**) MDCK cells expressing Gq-DREADD with GCaMP6s or Booster-PKA were mixed and plated, treated with 1 μM CNO, and imaged. Blue circled cells are producer cells, expressing Gq-DREADD. The FRET ratio, the value of mKate2/mKO $\kappa$, in each pixel is shown in pseudocolor as indicated. The time 0 was set as just before CNO addition. Cells showing an F/F0 value >4 were analyzed for their RSPA as *Figure 1C*. Red lines indicate their average value.

The online version of this article includes the following video for figure 3:

**Figure 3—video 1.** Correlation of calcium concentration with radial spread of PKA activation (RSPA).
https://elifesciences.org/articles/86727/figures#fig3video1

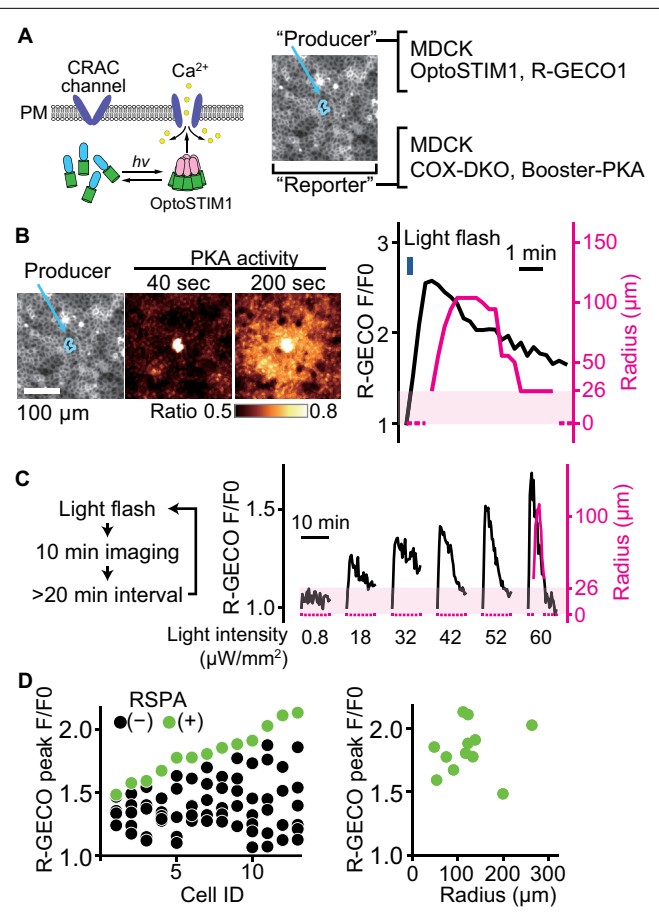

**Figure 4.** A switch-like response of radial spread of PKA activation (RSPA) to calcium transients. (**A**) Schematic representation of RSPA induction using OptoSTIM1. Madin–Darby canine kidney (MDCK) cells expressing OptoSTIM1 and R-GECO1 were employed as the producer cells. The Booster-PKA-expressing MDCK cells, deficient in COX-1 and COX-2 (COX-DKO), were employed as the reporter cells. (**B**) The producer cells were stimulated by a flashlight during imaging. Blue circled cells in the mKO$\kappa$ image are the producer cells. The FRET ratio, the value of mKate2/mKO$\kappa$, in each pixel is shown in pseudocolor as indicated. The time 0 was set as just before blue light irradiation. The detection limit for the RSPA radius was 26 μm, as shown in the shaded area. (**C**) Flashlight illumination was repeated with increasing LED power. (**D**) Vertically aligned dots (left) are the results from an individual producer cell. The right panel shows the relationship between R-GECO fluorescence intensity and the radius of RSPA.

dependence in the transcription of EP2, EP4 (*Figure 5D*), phosphodiesterases, or adenylyl cyclases (*Figure 5—figure supplement 1*). Thus, the cell density appears to increase the sensitivity to PGE$_2$ mostly in a transcription-independent manner.

## ERK activity is required for RSPA

Previously, we reported that ERK activation is propagated among confluent MDCK cells in a wave-like fashion (*Aoki et al., 2013*). To examine whether ERK activity also regulates RSPA, we simultaneously observed ERK and PKA activities by using EKAREV-NLS and Booser-PKA, respectively (*Figure 6A*). It appeared that the center of RSPA was localized primarily in areas of high ERK activity. Further quantitative analysis has shown that the ERK activity of the cells locating in the center of RSPA was significantly higher than that of the randomly chosen cells (*Figure 6B*, left). However, the size of RSPA did not correlate with the ERK activity (*Figure 6B*, right). We next examined the timing of RSPA and the passage of ERK activation waves by aligning the events at the highest PKA activity. It appears that RSPA was evoked when the cells exhibited the highest ERK activity (*Figure 6C and D*, *Figure 6—figure supplement 1*). Cross-correlation analysis of PKA activity and ERK activity revealed that ERK

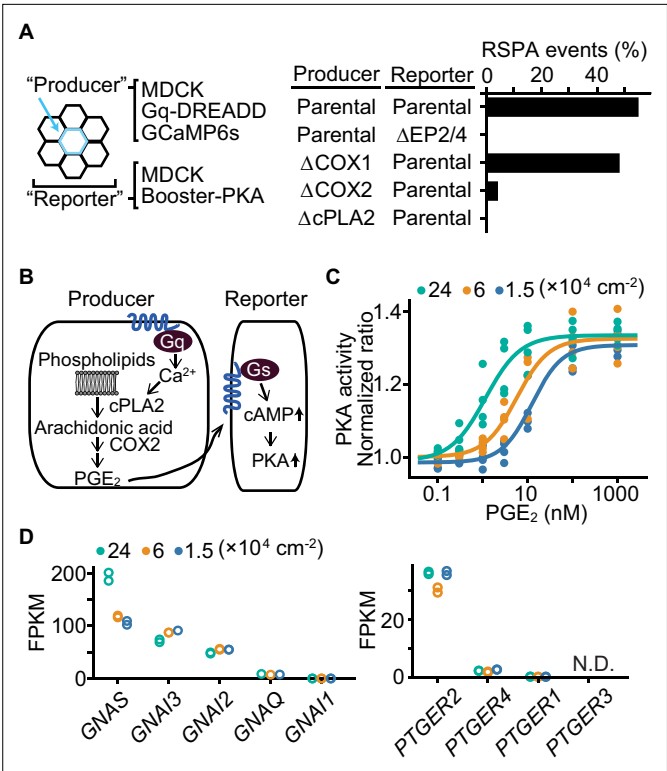

**Figure 5.** Effect of cell density on prostaglandin E₂ (PGE₂) sensitivity and the transcriptome. (**A**) Madin–Darby canine kidney (MDCK) cells expressing Gq-DREADD and GCaMP6s were employed as the parental producer cells. Meanwhile, MDCK cells expressing Booster-PKA were employed as the parental reporter cells. The genes knocked out by CRIPSR/Cas9 system are depicted in the figure. Analysis was performed as in *Figure 3G*. Each producer cell exhibiting F/F0 values >3 was analyzed for the occurrence of radial spread of PKA activation (RSPA). Data from two independent experiments was summed up. (**B**) Inter- and intracellular pathway of RSPA. (**C**) COX-DKO MDCK cells expressing Booster-PKA were plated at the indicated cell density, treated with increasing concentrations of PGE₂, and analyzed for PKA activity. The mKate2/mKO κ ratio representing PKA activity was calculated and plotted against PGE₂ concentration. The average intensity of the whole view field of mKate2 or mKO κ, at 20–30 min after the addition of PGE₂, was applied to calculate the mKate2/mKO κ ratio. Three or four independent experiments were performed. (**D**) COX-DKO MDCK cells were seeded at the indicated cell densities and subjected to RNA-seq analysis. FPKM values of other genes are in *Figure 5—figure supplement 1*. N.D. represents not detected.

The online version of this article includes the following figure supplement(s) for figure 5:

**Figure supplement 1.** Effect of cell density on the transcriptome.

activation preceded PKA activation by approximately 3 min (*Figure 6E*). The ERK activation wave is known to be mediated by EGFR and EGFR ligands (*Lin et al., 2022*). Accordingly, the addition of EGF faintly increased the frequency of RSPA in our experiments, while the MEK and EGFR inhibitors almost completely abrogated RSPA (*Figure 6F*), representing that ERK activation or basal ERK activity is essential for RSPA. Collectively, these results obtained with MDCK cells showed that RSPA is triggered by calcium transient in cells with high ERK activity.

The results in MDCK cells motivated us to validate our model in vivo. Thus, we tested RSPA in the basal layer of the mouse auricular epidermis could be canceled by the administration of MEKi (*Figure 6G and H*). As anticipated, RSPA in the basal layer was significantly attenuated 30 min after the administration, representing that ERK activity is required for RSPA in vivo.

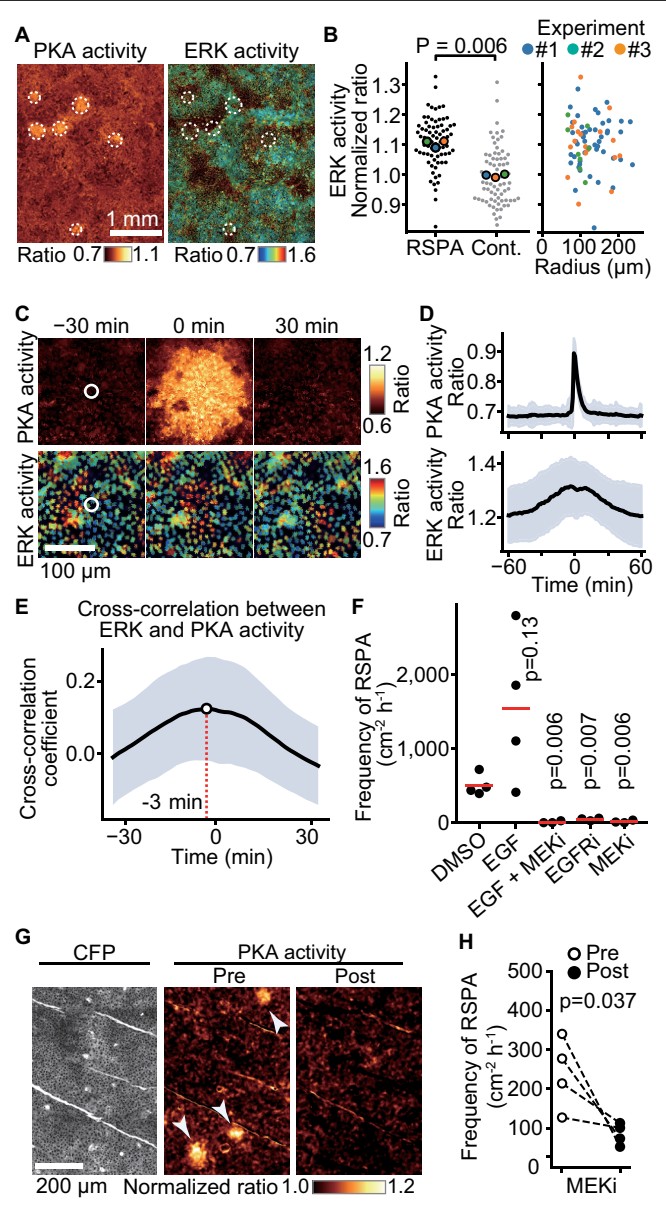

**Figure 6.** Requirement of ERK activation for radial spread of PKA activation (RSPA). (**A**) Madin–Darby canine kidney (MDCK) cells expressing EKAREV and Booster-PKA were observed for ERK and PKA activities every 5 min and 1 min, respectively. The mKate2/mKO$\kappa$ ratio image represents PKA activity in pseudocolor. The YPet/ECFP ratio image represents ERK activity in IMD mode. RSPA is indicated by white circles. (**B**) The ERK activities within 10 μm from the center of RSPA and within 10 μm from randomly set positions with a random number table generated by Python are plotted in the left panel. Each colored dot represents an average value of an independent experiment. The right scattered plot shows the relationship between ERK activity and the size of RSPA. (**C**) The correlation between ERK activation and RSPA is shown. This is a part of *Figure 6—video 1*. (**D**) Cross-correlation analysis of PKA and ERK activities. The average and SD values from 67 samples are shown in the black lines and blue shades, respectively. (**E**) Temporal cross-correlations between RSPA and ERK activation rate. The black line indicates the average temporal cross-correlation coefficients with SD. (**F**) MDCK cells expressing Booster-PKA were imaged in the presence of the following reagents: 0.1% v/v DMSO, 50 ng/mL EGF, 1 μM PD0325901 (MEKi), and 1 μM AG1478 (EGFRi). The frequency of RSPA was analyzed 20–80 min after the treatment. Each dot represents an individual experiment. Red lines indicate their average value. The control data set is from *Figure 1F*. p-Values were calculated between the labeled sample and the DMSO-treated sample. (**G, H**) Similar to *Figure 2*. Transgenic mice expressing AKAR3EV were observed under a two-photon excitation microscope and administrated a MEKi, 5 mg/kg PD0325901 intravenously. Shown are an ECFP image and a YPet/ECFP ratio image representing the cell

*Figure 6 continued on next page*

*Figure 6 continued*

density and PKA activity in pseudocolor, respectively. The images of PKA activity were projected over 30 min. The frequency of RSPA in pretreatment was quantitated more than 120 min before the injection. Similarly, the frequency of RSPA in a post-treatment was analyzed 15–90, 15–125, 15–85, and 15–115 min. Each dotted line represents an individual mouse experiment.

The online version of this article includes the following video and figure supplement(s) for figure 6:

**Figure supplement 1.** Representative ERK and PKA activities in the center of radial spread of PKA activation (RSPA).

**Figure supplement 2.** Effect of MEK inhibitor on calcium transients.

**Figure 6—video 1.** Correlation of ERK activity with radial spread of PKA activation (RSPA).
https://elifesciences.org/articles/86727/figures#fig6video1

**Figure 6—video 2.** Requirement of ERK activation for radial spread of PKA activation (RSPA) in vivo.
https://elifesciences.org/articles/86727/figures#fig6video2

## Discussion

### PGE$_2$ discharge causes radial spread of PKA activation in neighboring cells

To the best of our knowledge, only one study has visualized PGE$_2$ secretion from a single cell (*Zonta et al., 2003*). In this study, HEK cells expressing the Gq-coupled PGE$_2$ receptor EP1 and a calcium indicator were used to monitor PGE$_2$ release from agonist-stimulated astrocytes. However, the spontaneous release of PGE$_2$ has never been visualized in either tissue culture cells or live animals. Here we have shown that PGE$_2$ is discharged after calcium transients in MDCK cells at high cell densities (*Figure 1*). This PGE$_2$ discharge leads to the radial spread of PKA activation, which we named RSPA, in the neighboring EP2-expressing cells. By using transgenic mice expressing the PKA biosensor, RSPA was also observed in the mouse auricular epidermis (*Figure 2*). RSPA in the epidermis is almost completely shut off by COXi, strongly suggesting that prostaglandin(s), most likely PGE$_2$, mediates RSPA in the skin. Notably, we failed to observe RSPA in melanoma tissues in which calcium transients were frequently observed (*Konishi et al., 2021*). We reasoned that repetitive PGE$_2$ secretion from tumor cells maintains a high PGE$_2$ concentration in the tumor microenvironment, which prevented us from observing pulsatile PKA activation. In fact, the PGE$_2$ concentration in melanoma tissue is known to reach as high as 10 µM (*Konishi et al., 2021*). Thus, RSPA in the skin may function as an alert signal in an early phase of cellular stress.

### PGE$_2$ is discharged in a switch-like manner in response to Ca$^{2+}$ transients

Soon after the identification of a Ca$^{2+}$-dependent translocation domain within cPLA2 (*Clark et al., 1991*), Ca$^{2+}$-dependent cPLA2 arachidonic acid release from cells has been reported (*Hirabayashi et al., 1999*; *Gijón et al., 2000*); therefore, it is not surprising to find that the PGE$_2$ discharge in our present experiments was due to Ca$^{2+}$-dependent cPLA2 activation (*Figure 3*). However, visualization of PGE$_2$ secretion at the single-cell resolution revealed a switch-like response of PGE$_2$ discharge to the increasing Ca$^{2+}$ concentration (*Figure 4*). Recruitment of cPLA2 to the ER and perinuclear membrane requires a higher Ca$^{2+}$ concentration than that to Golgi (*Evans et al., 2001*). If so, cPLA2 may be sequestered at Golgi at low intracellular Ca$^{2+}$ concentration, and only when the intracellular Ca$^{2+}$ concentration exceeds the threshold that cPLA2 may reach the ER to liberate arachidonic acids.

What does regulate the frequency of Ca$^{2+}$ transients? Since the frequency is cell density-dependent in MDCK cells (*Figure 3D*), one candidate might be mechanical stress. 293 and HeLa cells demonstrate that cells show an increment in their frequency (*Morita et al., 2015*). However, a clear mechano receptor has not been identified.

### ERK also regulates the probability of RSPA

In the cell density-dependent RSPA of MDCK cells, ERK activity regulates the probability, but not the size, of RSPA. Of note, the MEK inhibitor did not significantly decrease the frequency of calcium transients (*Figure 6—figure supplement 2*), suggesting that ERK had a direct effect on the production

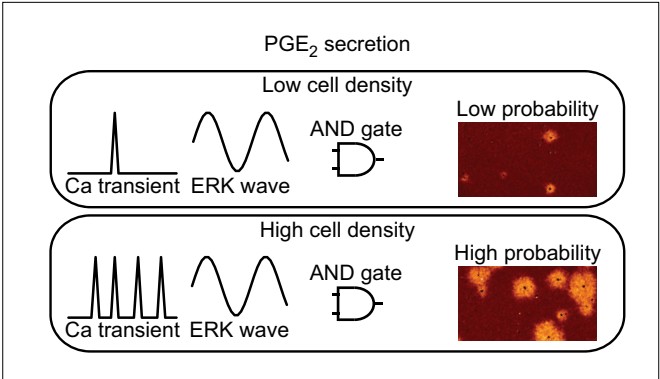

**Figure 7.** Models for prostaglandin E₂ (PGE₂) secretion. The frequency of calcium transients is cell density-dependent manner. The ERK activation wave is there in both conditions. Because both calcium transient and ERK activation are required for radial spread of PKA activation (RSPA), the probability for PGE₂ secretion is regulated as 'AND gate'.

of PGE₂. Because ERK positively regulates cPLA2 by phosphorylating Ser[505] (*Cook and McCormick, 1993*; *Qiu et al., 1993*), it is reasonable that the RSPA is regulated by ERK activity.

The ERK activation wave is operated by a positive feedforward mechanism in which ERK promotes EGFR ligand shedding and the following EGFR activation increases ERK activity in the neighboring cells (*Hino et al., 2020*; *Aoki et al., 2017*). Here we found that the ERK activation functions as the 'AND gate' for PGE₂ production together with the calcium transients (*Figure 7*). Notably, the propagation of PKA activation, ~100 µm/min (*Figure 1B*), is markedly faster than that of ERK activation, 2–4 µm/min (*Hiratsuka et al., 2015*). Because PKA antagonizes Ras-dependent ERK activation (*Cook and McCormick, 1993*; *Burgering et al., 1993*; *Wu et al., 1993*), the EGFR ligand-ERK and PGE₂-PKA pathways fit the Turing diffusion reaction model consisting of slow positive and fast negative signaling cascades.

## RSPA may not directly affect cell competition

Recently, PGE₂ was shown to regulate cell competition among MDCK cells. Interestingly, extrusion of Ras-transformed MDCK cells has been shown to be suppressed by PGE₂ (*Sato et al., 2020*), whereas extrusion of MDCK cells expressing constitutively active YAP was dependent on PGE₂ (*Ishihara et al., 2020*). Therefore, the effect of PGE₂ in cell competition could be markedly different according to the signaling cascades that cause the oncogenic changes of MDCK cells. Importantly, PGE₂ promotes the extrusion of MDCK cells expressing the constitutively active YAP by internalization of E-cadherin (*Ishihara et al., 2020*), which is a relatively slow process. Since RSPA causes PKA activation for only several minutes in each cell, multiple RSPA events may be needed to reach the concentration required for the induction of extrusion.

## The radius of RSPA might represent the amount of PGE₂ discharge

The radius of RSPA, in MDCK cells, might partially represent the amount of PGE₂ discharge from the single cell. However, because of the cell-to-cell junction of the confluent MDCK cells, we failed to quantify the PGE₂. In HeLa cells, we quantitated PGE₂ secreted from a single cell by combining fluorescence microscopy and a simulation model (*Watabe et al., 2023*).

## Limitations of this study

It would be clear that the physiological relevance of RSPA remains. In MDCK cells, the treatment of NSAIDs, which perturb RSPA, did not show a detectable change in cell growth, ERK wave propagation during collective migration, migration velocity, cell survival, or apoptosis. In mice epidermis, the frequency of RSPA was not remarkably changed by inflammation or corrective migration, evoked by TPA treatment or wound, respectively.

## Conclusions

We have shown that the $PGE_2$ discharge from a single cell is a stochastic and switch-like event in the confluent MDCK cells and mouse epidermis. The secreted $PGE_2$ can transiently activate PKA in cells within a few hundred micrometers from the producer cell. The question of why cells adopt this pulsatile rather than continuous secretion of $PGE_2$ awaits the future.

# Materials and methods

## Reagents

ONO-AE3-208, PF-04418948, clozapine N-oxide, and $PGE_2$ were purchased from Cayman Chemical. BAPTA-AM was obtained from Enzo Life Sciences. PD0325901, mitomycin C, and indomethacin were purchased from FUJIFILM Wako Pure Chemical Corp. AG1478 was purchased from BioVision Inc. Apyrase and EGF were obtained from Sigma-Aldrich. H-89 was purchased from Seikagaku Corp. The DREADD ligand, clozapine N-oxide, was purchased from Cayman Chemical. Flurbiprofen axetil was purchased from KAKEN Pharmaceutical.

## Cell culture

MDCK cells were purchased from the RIKEN BioResource Center (no. RCB0995). Lenti-X 293T cells were obtained from Invitrogen. MDCK and Lenti-X 293T cells were maintained in Dulbecco's modified Eagle medium (FUJIFILM Wako Pure Chemical Corp.) containing 10% fetal bovine serum (Sigma-Aldrich) and 1% v/v penicillin−streptomycin (Nacalai Tesque). Cell line identity were validated.

## Plasmids and primers

Plasmids and primers are described in *Supplementary files 1 and 2*.

## Cell lines

For the generation of MDCK cells stably expressing Booster-PKA or the other ectopic proteins, a lentiviral or piggyBac transposon system was employed. To prepare the lentivirus, a lentiCRISPRv2-derived expression plasmid, psPAX2 (plasmid no. 12260; Addgene), and pCMV-VSV-G-RSV-Rev (RIKEN BioResource Center) were co-transfected into Lenti-X 293T cells using polyethyleneimine (Polyscience). Virus-containing media were collected at 48 or 72 hr after transfection, filtered, and applied to target cells with 10 µg/mL polybrene (Nacalai Tesque). To introduce ectopic genes using a PiggyBac system, pPB plasmids and pCMV-mPBase(neo-) encoding piggyBac transposase were co-transfected into MDCK cells by electroporation with an Amaxa nucleofector (Lonza). Cells were selected with the medium containing the following antibiotics: 10 µg/mL blasticidin S (FUJIFILM Wako Pure Chemical Corp.), 100 µg/mL zeocin (InvivoGen), 2.0 µg/mL puromycin (InvivoGen), or 200 µg/mL hygromycin (FUJIFILM Wako Pure Chemical Corp.).

MDCK cells expressing EKAREV-NLS were previously described (*Kawabata and Matsuda, 2016*). The established cell lines are described in *Supplementary file 3*.

The identity of cell lines was validated by partial genome sequence during knockout cell development. Mycoplasma contamination was tested with PlasmoTest (InvivoGen) when it was suspicious.

## CRISPR/Cas9-mediated KO cell lines

For CRISPR/Cas9-mediated single or multiple knockouts of genes, sgRNAs targeting the exons were designed using CRISPRdirect (*Naito et al., 2015*). Oligo DNAs for the sgRNA were cloned into the lentiCRISPRv2 (plasmid no. 52961; Addgene) vector or pX459 (plasmid no. 62988; Addgene) vector. The expression plasmids for sgRNA and Cas9 were introduced into MDCK cells by lentiviral infection or electroporation. For electroporation, pX459-derived plasmids were transfected into MDCK cells using an Amaxa Nucleofector II. Cells were selected with the medium containing the antibiotics depending on the drug-resistance genes. After the selection, genomic DNAs were isolated with SimplePrep reagent (TaKaRa Bio). PCR was performed using KOD FX neo (Toyobo) for amplification with the designed primers, followed by DNA sequencing.

## Wide-field fluorescence microscopy

Cells were imaged with an ECLIPSE Ti2 inverted microscope (Nikon) or an IX83 inverted microscope (Olympus). The ECLIPSE Ti2 inverted microscope was equipped with a Plan Fluor ×10 or ×4 objective,

an ORCA Fusion Digital CMOS camera (HAMAMATSU PHOTONICS K.K.), an X-Cite TURBO LED light source (Excelitas Technologies), a Perfect Focus System (Nikon), a TI2-S-SE-E motorized stage (Nikon), and a stage top incubator (Tokai Hit). The IX83 inverted microscope was equipped with a UPlanAPO ×10/0.40 NA objective lens (Olympus), a Prime sCMOS camera (Photometrics), a CoolLED preci-sExcite LED illumination system (Molecular Devices), an IX2-ZDC laser-based autofocusing system (Olympus), and an MD-XY30100T-Meta automatically programmable XY stage.

The following filters were used for the multiplexed imaging: for CFP and YFP imaging, a 434/32 excitation filter (Nikon), a dichroic mirror 455 (Nikon), and 480/40 and 535-30 emission filters (Nikon) for CFP and YFP, respectively; for GCaMP6s imaging, a 480/40 (Nikon) excitation filter, a dichroic mirror 455 (Nikon), and a 535/50 emission filter (Nikon); for mKOκ and mKate2 imaging, a 555BP10 excitation filter (Omega Optical), an FF562Di03 dichroic mirror (Semrock), and XF3024 (590DF35) (Omega Optical) and BLP01-633R-25 (Semrock) emission filters for mKOκ and mKate2, respectively; for iRFP670 imaging, an FF01-640/14 excitation filter (Semrock), a dichroic mirror 660 (Nikon), and a 700/75 emission filter (Nikon); for R-GECO1, a 555BP10 excitation filter (Omega Optical), an FF562Di03 dichroic mirror (Semrock), and an XF3024 emission filter (590DF35) (Omega Optical).

## In vivo two-photon imaging of the mouse epidermis

The establishment of transgenic mice expressing AKAR3EV (PKAchu mice) was described previously (*Kamioka et al., 2012*). Briefly, 8- to 13-week-old female mice were used for the in vivo imaging. The ear hair was removed with a razor 1 d before the experiments. Mice were anesthetized with 1.5% isoflurane (FUJIFILM Wako Pure Chemical Corp.) inhalation and placed in a side-lying position on an electric heater maintained at 37°C. The ear skin was placed on the cover glass. Two-photon exci-tation microscopy was performed with an FV1200MPE-IX83 inverted microscope (Olympus) equipped with a ×30/1.05 silicon oil-immersion objective lens (XLPLN 25XWMP; Olympus), an InSight DeepSee Ultrafast laser (Spectra Physics), an IR-cut filter (BA685RIF-3), two dichroic mirrors DM505 (Olympus), and two emission filters (BA460-500 for CFP and BA520-560 for YFP) (Olympus). The excitation wave-length was 840 nm.

The animal protocols were approved by the Animal Care and Use Committee of Kyoto University Graduate School of Medicine (approval no. 22063).

## Spontaneous RSPA

MDCK cells expressing Booster-PKA or hyBRET-Epac were seeded on collagen-coated glass-bottom 96-well plates (Matsunami Glass Ind.) at a density of $1.2–2.4 \times 10^5$ cells/cm$^2$. Before imaging, the culture media were replaced with phenol red-free M199 (Thermo Fisher Scientific) supplemented with 10% fetal bovine serum. Cells were imaged by wide-field fluorescence microscopy, as described above.

## Analysis of calcium concentrations

Intracellular Ca$^{2+}$ concentrations in MDCK cells were visualized with a genetically encoded calcium indicator, GCaMP6s or R-GECO1. For GCaMP6s analysis, calcium signals were expressed as F/F0, where F is the fluorescence at each time point, and F0 represents baseline fluorescence. To analyze the peak F/F0 value of GCaMP6s, F0 was calculated as the minimum projection of fluorescence inten-sity over the 5 min before each frame. Each cell showing calcium transient was visually checked to exclude the F/F0 elevation caused by flowing debris and misregistration of cells. If two or more adja-cent cells showed calcium transients simultaneously, it was counted as a calcium transient. If two or more calcium transients were detected at intervals of more than 1 min, they were counted separately. In the Gq-DREADD experiment, F0 was calculated as the mean intensity before the stimulation.

For R-GECO1 analysis, F/F0 calcium signals were calculated by assigning the reference F0 using the fluorescence intensity before each blue light flash.

## Gq-DREADD-induced calcium transients and RSPA

MDCK cells expressing both Gq-DREADD-P2A-mCherry-NLS (*Evans et al., 2001*) and GCaMP6s (*Evans et al., 2001*) were utilized as PGE$_2$ producer cells. MDCK cells expressing Booster-PKA served as PGE$_2$ reporter cells. The producer and reporter cells were mixed at a ratio of 1:400 to 1:200 and plated on collagen-coated glass-bottom 96-well plates (Matsunami Glass Ind. or AGC Inc) at a density

of 1.2–2.4 × $10^5$ cells/cm$^2$. Before imaging, the culture media were replaced with phenol red-free M199 (Thermo Fisher Scientific) supplemented with 10% fetal bovine serum. Gq-DREADD was activated by the addition of 1 µM of clozapine N-oxide (CNO). Cells were imaged by wide-field fluorescence microscopy, as described above.

## Light-induced calcium transients and RSPA

MDCK cells expressing both R-GECO1-P2A-iRFP670 and OptoSTIM1 (CRY2clust) (*Lee et al., 2014*) were used as PGE$_2$ producer cells. COX1 and COX2-deficient MDCK cells expressing Booster-PKA were used as PGE$_2$ reporter cells. The producer and reporter cells were mixed at a ratio of 1:400 to 1:1200 and plated on collagen-coated glass-bottom 96-well plates (Matsunami Glass Ind.) or 24-well plates (AGC Inc). After 16–32 hr of incubation, the culture media were replaced with phenol red-free M199 (Thermo Fisher Scientific) supplemented with 10% fetal bovine serum or M199 supplemented with 0.1% w/v bovine serum albumin (Sigma-Aldrich). Cells were imaged by wide-field fluorescence microscopy, as described above. During the observation, OptoSTIM1 was activated with 475 nm LED for 200 ms to trigger calcium influx into the cell. To control the calcium influx from small to large, the excitation light was modulated from 0.8 to 67 µW mm$^{-2}$. To prevent cell division, MDCK cells with 3 µg/mL of mitomycin C for 1 hr 1 d before passage.

## Titration of PGE$_2$ sensitivity

COX-1 and COX-2 depleted (COX-DKO) MDCK cells expressing Booster-PKA were seeded on a 96-well glass-base plate at the indicated densities. Before imaging, the culture media were replaced with phenol red-free M199 (Thermo Fisher Scientific) supplemented with 10% fetal bovine serum for MDCK. The 96-well plate was imaged by an inverted microscope as described earlier. mKate2 and mKOκ images were obtained in one position for every well at around 5 min intervals. Cells were stimulated with PGE$_2$ at the indicated concentrations. The mKate2/mKOκ ratio was quantified from the average intensity of the whole field of view at around 20–30 min after the addition of PGE$_2$.

## Quantification of RSPA

The program code for image analysis is available via GitHub at https://github.com/TetsuyaWatabe-1991/RSPAanalysis (copy archived at *Watabe, 2023*).

Ratio images of MDCK cells expressing Booster-PKA were created after background subtraction. A median filter and a Gaussian 2D filter were applied to each image for noise reduction. The ratio image was normalized by a minimum intensity projection along the time axis. The processed images were binarized with a predetermined threshold and processed by morphological opening and closing to refine the RSPA area. Center coordinates and equivalent circle radii were obtained from each RSPA area. If the distance between the center coordinates of RSPA between successive frames was less than 100 µm, they were considered to be the same RSPA. For MDCK cells expressing hyBRET-Epac and mouse ear skin expressing AKAR3EV, the center coordinates of each RSPA were manually determined due to the low signal-to-noise ratio.

To obtain the time course of the RSPA radius, concentric regions were defined at the center of each RSPA. The median FRET ratio in each concentric ROI was calculated. The radius of the outermost concentric region where the median ratio value exceeds a predetermined threshold was defined as the radius of the RSPA.

## Cross-correlation analysis of ERK and PKA activity

Cross-correlation analysis was performed with Python using the scientific library SciPy (http://www.scipy.org). The centers of each spontaneous RSPA in MDCK cells expressing both EKAREV and Booster-PKA were detected automatically as described above. The regions of interest were defined at the center of each RSPA with a radius of 10 µm, and the average ERK and PKA activity from 2 to 5 cells was quantified. The program code for this analysis is available via GitHub as described above.

## RNA-seq

COX1 and COX2-deficient MDCK cells expressing Booster-PKA were seeded in collagen-coated glass-bottom 96-well plates (AGC Inc) at a density of 1.5 × $10^4$ or 6.0 × $10^4$ or 2.4 × $10^5$ cells/cm$^2$. After 24 hr of incubation, the culture media were replaced with phenol red-free M199 (Thermo Fisher

Scientific) supplemented with 10% fetal bovine serum. Then, 3 hr after medium replacement, RNA was extracted from each sample using an RNeasy Mini Kit (QIAGEN). Libraries for RNA-seq were prepared using an NEBNext Ultra II Directional RNA Library Prep Kit for Illumina (New England Biolabs) and sequenced on the NextSeq500 (Illumina) as 75 bp single-end reads. RNA-seq data were trimmed using Trim Galore version 0.6.6 (*Krueger, 2015Krueger, 2015*) and Cutadapt version 2.8 (*Martin, 2011*). The quality of reads was checked and filtered using FastQC version 0.11.9 (; *Andrews, 2010*). The reads were mapped to a reference genome canFam3.1 (*Lindblad-Toh et al., 2005*, *Hoeppner et al., 2014*) using HISAT2 version 2.2.1 (*Kim et al., 2019*), and the resulting aligned reads were sorted and indexed using SAMtools version 1.7 (*Li et al., 2009*). Relative abundances of genes were measured in FPKM using StringTie version 2.1.4 (*Kovaka et al., 2019*, *Pertea et al., 2015*). Plots were created in Python using the pandas, matplotlib, NumPy, and seaborn libraries.

Sequence data are available in the DNA Data Bank of Japan Sequence Read Archive under accession numbers DRR014156 to DRR014161.

## Statistical analysis

All statistical analyses and visualizations were performed in Python using the libraries NumPy, pandas, SciPy, pingouin, matplotlib, and seaborn. No statistical analysis was used to predetermine the sample size. Welch's *t*-test was used to evaluate statistically significant differences. p-Values <0.05 were considered statistically significant.

## Acknowledgements

We are grateful to the members of the Matsuda Laboratory for their helpful input, K Hirano, T Uesugi, and Y Takeshita, who provided technical assistance, and to the Medical Research Support Center of Kyoto University for DNA sequence analysis. We thank Takefumi Kondo and Yukari Sando (NGS core facility of the Graduate Schools of Biostudies, Kyoto University) for supporting the RNA-seq analysis. This work was supported by the Kyoto University Live Imaging Center. Financial support was provided by JSPS KAKENHI grants (nos. 21K20773 to TW, 21H02715 and 21H05226 to KT, and 19H00993 and 20H05898 to MM), a JST CREST grant (no. JPMJCR1654 to MM), a JST Moonshot R&D grant (no. JPMJPS2022-11 to MM), a grant from Fugaku Foundation (to MM), and a grant from Research Grant of the Princess Takamatsu Cancer Research Fund (to KT).

## Additional information

### Funding

| Funder | Grant reference number | Author |
| --- | --- | --- |
| Japan Society for the Promotion of Science | 21K20773 | Tetsuya Watabe |
| Japan Society for the Promotion of Science | 21H02715 | Kenta Terai |
| Japan Society for the Promotion of Science | 21H05226 | Kenta Terai |
| Japan Society for the Promotion of Science | 19H00993 | Michiyuki Matsuda |
| Japan Society for the Promotion of Science | 20H05898 | Michiyuki Matsuda |
| Core Research for Evolutional Science and Technology | JPMJCR1654 | Michiyuki Matsuda |
| Moonshot Research and Development Program | JPMJPS2022-11 | Michiyuki Matsuda |
| Fugaku Foundation | | Michiyuki Matsuda |

| Funder | Grant reference number | Author |
|---|---|---|
| Princess Takamatsu Cancer Research Fund | | Kenta Terai |

The funders had no role in study design, data collection and interpretation, or the decision to submit the work for publication.

## Author contributions

Tetsuya Watabe, Conceptualization, Resources, Data curation, Formal analysis, Funding acquisition, Validation, Investigation, Methodology, Writing - original draft; Shinya Yamahira, Kanako Takakura, Investigation; Dean Thumkeo, Shuh Narumiya, Writing - review and editing; Michiyuki Matsuda, Resources, Supervision, Funding acquisition, Project administration, Writing - review and editing; Kenta Terai, Conceptualization, Resources, Data curation, Formal analysis, Supervision, Funding acquisition, Validation, Investigation, Visualization, Methodology, Project administration, Writing - review and editing

## Author ORCIDs

Michiyuki Matsuda (iD) http://orcid.org/0000-0002-5876-9969
Kenta Terai (iD) http://orcid.org/0000-0001-7638-3720

## Ethics

The animal protocols were approved by the Animal Care and Use Committee of Kyoto University Graduate School of Medicine (approval nos. 22063).

Reviewer #1 (Public Review): https://doi.org/10.7554/eLife.86727.3.sa1
Reviewer #2 (Public Review): https://doi.org/10.7554/eLife.86727.3.sa2
Author Response https://doi.org/10.7554/eLife.86727.3.sa3

# Additional files

## Supplementary files

- MDAR checklist
- Supplementary file 1. Plasmids in this paper.
- Supplementary file 2. Primers for validating the gene knockout.
- Supplementary file 3. Cell lines.

## Data availability

Sequence data are available in the DNA Data Bank of Japan Sequence Read Archive under accession numbers DRA014156 to DRA014161.

The following datasets were generated:

| Author(s) | Year | Dataset title | Dataset URL | Database and Identifier |
|---|---|---|---|---|
| Watabe T, Yamahira S, Takakura K, Thumkeo D, Narumiya S, Matsuda M, Terai K | 2023 | Density-dependent PGE2 sensitivity in MDCK | https://ddbj.nig.ac.jp/resource/sra-submission/DRA014156 | DNA Data Bank of Japan Sequence Read Archive, DRA014156 |
| Watabe T, Yamahira S, Takakura K, Thumkeo D, Narumiya S, Matsuda M, Terai K | 2023 | Density-dependent PGE2 sensitivity in MDCK | https://ddbj.nig.ac.jp/resource/sra-submission/DRA014157 | DNA Data Bank of Japan Sequence Read Archive, DRA014157 |
| Watabe T, Yamahira S, Takakura K, Thumkeo D, Narumiya S, Matsuda M, Terai K | 2023 | Density-dependent PGE2 sensitivity in MDCK | https://ddbj.nig.ac.jp/resource/sra-submission/DRA014158 | DNA Data Bank of Japan Sequence Read Archive, DRA014158 |

*Continued on next page*

*Continued*

| Author(s) | Year | Dataset title | Dataset URL | Database and Identifier |
|---|---|---|---|---|
| Watabe T, Yamahira S, Takakura K, Thumkeo D, Narumiya S, Matsuda M, Terai K | 2023 | Density-dependent PGE2 sensitivity in MDCK | https://ddbj.nig.ac.jp/resource/sra-submission/DRA014159 | DNA Data Bank of Japan Sequence Read Archive, DRA014159 |
| Watabe T, Yamahira S, Takakura K, Thumkeo D, Narumiya S, Matsuda M, Terai K | 2023 | Density-dependent PGE2 sensitivity in MDCK | https://ddbj.nig.ac.jp/resource/sra-submission/DRA014160 | DNA Data Bank of Japan Sequence Read Archive, DRA014160 |
| Watabe T, Yamahira S, Takakura K, Thumkeo D, Narumiya S, Matsuda M, Terai K | 2023 | Density-dependent PGE2 sensitivity in MDCK | https://ddbj.nig.ac.jp/resource/sra-submission/DRA014161 | DNA Data Bank of Japan Sequence Read Archive, DRA014161 |

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
