## [Editor Report · eLife assessment]

This **important** study reports on the dynamics of PKA investigated at the single-cell level in vitro and in epithelia in vivo. Using different fluorescent biosensors and optogenetic actuators, the authors dissect the signaling pathway responsible for PKA waves, finding that PKA activation is a consequence of PGE_2_ release, which in turn is triggered by calcium pulses, requiring high ERK activity. The evidence supporting the claims is **solid**. At this stage, the work is still partly descriptive in nature, and additional measurements would increase the strength of mechanistic insights and physiological relevance.

---

## [Referee Report · Reviewer #1 (Public Review)]

This research article by Watabe T and colleagues characterizes PKA waves triggered by prostaglandin E2 (PGE2). What the author discovered is that waves of PKA occur both in vitro, in MDCK epithelial monolayers, and in vivo, in the ear epidermis in mice. The PKA waves are the consequence PGE2 discharge, that in turn is triggered by Calcium bursts. Calcium level and ERK activity intensity control that mechanism by acting at different levels.

This article is a technological tour de force using different biosensors and optogenetic actuators. However, what makes this article interesting is the ability of combining these tools together to dissect a complex signaling pathway at the single-cell level and with highly dynamic processes. For this reason, this paper represents the essence of modern cell biology and paves the way for the cell biology of the future.

However, we think that the paper in this stage is still partly descriptive in its nature, and more measurements are needed to increase the strength of the mechanistic insights. Here below the points that we believe that need some improvement.

1)Even though the phenomenon of PGE2 signal propagation is elegantly demonstrated and well described, the whole paper is mostly of descriptive nature - the PGE2 signal is propagated via intercellular communication and requires Ca transients as well as MAPK activity, however function of these RSPAs in dense epithelium is not taken into consideration.

What is the function of these RSPAs in cellular crowding? - Does it promote cell survival or initiate apoptosis? Does it feed into epithelial reorganization during cellular crowding? Still something else? The authors discuss possible roles of this phenomenon in cell competition context, but show no experimental or statistical efforts to answer this question. I believe some additional analysis or simple experiment would help to shed some light on the functional aspect of RSPAs and increase the importance of all the elegant demonstrations and precise experimental setups that the manuscript is rich of. Monolayer experiments using some perturbations that challenge the steady state of epithelial homeostasis - drug treatments/ serum deprivation/ osmotic stress/ combined with live cell imaging and statistical methods that take into account local cell density might provide important answers to these questions. The authors could consider following some of these ideas to improve the overall value of the manuscript.

1. In the line 82-84 the authors claim: "We found that the pattern of cAMP concentration change is very similar to the activity change of PKA, indicating that a Gs protein-coupled receptor (GsPCR) mediates RSPA". In our opinion, this conclusion is not well-supported by the results. The authors should at least show that some measurement of the two patterns show correlation. Are the patterns of cAMP of the same size as the pattern of PKA? Do they have the same size depending on cell density? Do they occur at the same frequency as the PKA patterns, depending on the cell density? Do they have an all or nothing activation as PKA or their activation is shading with the distance from the source?

2. In general, the absolute radius of the waves is not a good measurement for single-cell biology studies, especially when comparing different densities or in vivo vs in vitro experiments. We suggest the authors to add the measurement of the number of the cells involved in the waves (or the radius expressed in number of cells).

3. In 6D, the authors should also show the single-cell trajectories to understand better the correlation between PKA and ERK peaks. Is the huger variability in ERK activity ratio dues to different peak time or different ERK activity levels in different cells? The authors should show both the variability in the time and intensity.

4. In lines 130-132, the authors write, "This observation indicates that the amount of PGE2 secretion is predetermined and that there is a threshold of the cytoplasmic calcium concentration for the triggered PGE2 secretion". How could the author exclude that the amount of PGE2 is not regulated in its intensity as well? For sure, there is a threshold effect regarding calcium, but this doesn't mean that PGE2 secretion can be further regulated, e.g. by further increasing calcium concentration or by other mechanisms.

5. The manuscript shows that not all calcium transients are followed by RSPAs. Does the local cell density/crowding increase the probability of overlap between calcium transients and RSPAs?

The revision of the Watabe T paper provides additional data and analyses in response to the reviewers' comments. On our side, we are satisfied by these improvements.

In the answer to our first question, the authors claim that they did multiple experiments to understand the function of RSPA in MDCK cell, all providing negative results. The authors could consider publishing the negative results as well, as they can be useful for the community.

In sum, we are convinced of the value of this article, and we thank the authors for the work that has been done.

---

## [Referee Report · Reviewer #2 (Public Review)]

This study visualizes a specific localized form of cell-to-cell communication and conveys very well with what dynamics and sensitivity this biological phenomenon occurs.

Using a FRET-based PKA biosensor, the authors observed that radial localized kinase activity changes spontaneously occur in adjacent cells of certain cell density. This phenomenon of radial propagation of PKA activity changes in groups of cells was further mechanistically elucidated and characterized. Interestingly, the authors found that individual cells in the cell groups form spontaneous Ca2+ transients, which at a certain strength can trigger the biosynthesis and release of prostaglandin E2 (PGE2). PGE2 then acts on the neighboring cells and triggers the increase of cAMP levels and the associated activation of the PKA via G-protein-coupled receptors (EP2 and EP4). In systematic, well-structured experiments, it was then found that the frequency of occurrence of such radial activations depends not only on the cell density but also on the activation state of the ERK MAP kinase pathway.

Strength

In this study, the authors skillfully used various modern genetically encoded biosensors and other tools such as optogenetic tools to visualize and characterize an interesting biological phenomenon of cell-to-cell communication. The insights gained with these investigations produce a better understanding of the dynamics, sensitivity, and spatial extent with which such communications can occur in a cell network. It is also worth noting that the authors have not limited the studies to 2D cell culture in vitro, but were also able to confirm the findings in an animal model.

Weakness

The work is hardly conclusive as to the actual biological significance of the phenomenon. It would be interesting to know more under which physiological and pathological conditions PGE2 triggers such radical PKA activity changes. It is not well explained in which tissues and organs and under what conditions this type of cell-to-cell communication could be particularly important.

The authors also do not explain further why in certain cells of the cell clusters Ca2+ signals occur spontaneously and thus trigger the phenomenon. What triggers these Ca2+ changes? And why could this be linked to certain cell functions and functional changes?

What explains the radius and the time span of the radial signal continuation? To what extent are these factors also related to the degradation of PGE2? The work could be stronger if such questions and their answers would be experimentally integrated and discussed.

---

## [Author Response]

The following is the authors’ response to the original reviews.

**eLife assessment**
This important study reports investigation of the dynamics of PKA at the single-cell level in in vitro and in epithelia in vivo. Using different fluorescent biosensors and optogenetic actuators, the authors dissect the signaling pathway responsible for PKA waves, finding that PKA activation is a consequence of PGE2 release, which in turn is triggered by calcium pulses, requiring high ERK activity. The evidence supporting the claims is solid. At this stage the work is still partly descriptive in nature, and additional measurements would increase the strength of mechanistic insights and physiological relevance.

We deeply appreciate Dr. Alejandro San Martín and Dr. Jonathan Cooper and the reviewers. Each comment is valuable and reasonable. We will revise our paper as much as possible.

We have described what we will do for the reviewer’s comments one by one in the below section.

**Reviewer #1 (Recommendations For The Authors):**
1. Even though the phenomenon of PGE2 signal propagation is elegantly demonstrated and well described, the whole paper is mostly of descriptive nature - the PGE2 signal is propagated via intercellular communication and requires Ca transients as well as MAPK activity, however function of these RSPAs in dense epithelium is not taken into consideration. What is the function of these RSPAs in cellular crowding? - Does it promote cell survival or initiate apoptosis? Does it feed into epithelial reorganization during cellular crowding? Still something else? The authors discuss possible roles of this phenomenon in cell competition context, but show no experimental or statistical efforts to answer this question. I believe some additional analysis or simple experiment would help to shed some light on the functional aspect of RSPAs and increase the importance of all the elegant demonstrations and precise experimental setups that the manuscript is rich of. Monolayer experiments using some perturbations that challenge the steady state of epithelial homeostasis - drug treatments/ serum deprivation/ osmotic stress/ combined with live cell imaging and statistical methods that take into account local cell density might provide important answers to these questions. The authors could consider following some of these ideas to improve the overall value of the manuscript.

We would like to thank the reviewer’s comment. Although we have intensively tried to identify the physiological relevance of RSPA, we could not detect the function at present.

In the case of MDCK, the treatment of NSAIDs, which cancels RSPA, did not affect its cell growth, ERK wave propagation during collective migration, migration velocity, cell survival, or apoptosis. In mouse epidermis, the frequency of RSPA was NOT affected by inflammation and collective cell migration, evoked by TPA treatment and wound, respectively.

Notably, RSPA also occurs in the normal epidermis, implying its relevance in homeostasis. However, at the current stage, we believe that the PGE2 dynamics and its regulation mechanism in the normal epidermis would be worth reporting to researchers in the field.

1. In the line 82-84 the authors claim: "We found that the pattern of cAMP concentration change is very similar to the activity change of PKA, indicating that a Gs protein-coupled receptor (GsPCR) mediates RSPA". In our opinion, this conclusion is not well-supported by the results. The authors should at least show that some measurements of the two patterns show correlation. Are the patterns of cAMP of the same size as the pattern of PKA? Do they have the same size depending on cell density? Do they occur at the same frequency as the PKA patterns, depending on the cell density? Do they have an all or nothing activation as PKA or their activation is shading with the distance from the source?

We have modified the text (line85)

“Although the increment of the FRET ratio was not so remarkable as that of Booster-PKA, Wwe found that the pattern of cAMP concentration change is very similar to the activity change of PKA, indicating that a Gs protein-coupled receptor (GsPCR) mediates RSPA. This discrepancy may be partially explained by the difference in the dynamic ranges for cAMP signaling in each FRET biosensor (Watabe2020). “

1. In general, the absolute radius of the waves is not a good measurement for single-cell biology studies, especially when comparing different densities or in vivo vs in vitro experiments. We suggest the authors add the measurement of the number of the cells involved in the waves (or the radius expressed in number of cells).

We appreciate the reviewer’s comment. We have analyzed our results to demonstrate the number of cells as in Fig2E, which would be easy for readers to understand.

1. In 6D, the authors should also show the single-cell trajectories to understand better the correlation between PKA and ERK peaks. Is the huger variability in ERK activity ratio dues to different peak time or different ERK activity levels in different cells? The authors should show both the variability in the time and intensity.

We have added a few representative results as Fig. S4.

1. In lines 130-132, the authors write, "This observation indicates that the amount of PGE2 secretion is predetermined and that there is a threshold of the cytoplasmic calcium concentration for the triggered PGE2 secretion". How could the author exclude that the amount of PGE2 is not regulated in its intensity as well? For sure, there is a threshold effect regarding calcium, but this doesn't mean that PGE2 secretion can be further regulated, e.g. by further increasing calcium concentration or by other mechanisms.

We agree with the reviewer’s comment. We have modified the text.

1. The manuscript shows that not all calcium transients are followed by RSPAs. Does the local cell density/crowding increase the probability of overlap between calcium transients and RSPAs?

We appreciate the reviewer’s comment. We have also hypothesized the model. However, we did not see the correlation that the reviewer pointed out. Currently, the increment of the RSPA frequency at high density is partially caused by the increment of calcium transients.

**Reviewer #2 (Recommendations For The Authors):**
1. The work is hardly conclusive as to the actual biological significance of the phenomenon. It would be interesting to know more under which physiological and pathological conditions PGE2 triggers such radial PKA activity changes. It is not well explained in which tissues and organs and under what conditions this type of cell-to-cell communication could be particularly important.The greatest weakness of the study seems to be that the biological significance of the phenomenon is not clearly clarified. Although it can be deduced that PKA activation has many implications for cell signaling and metabolism, the work lacks the actual link to physiological or pathological significance.

We deeply appreciate the reviewer’s comment. Similar to the reseponse of reviewer#1, although we have intensively tried to identify the physiological relevance of RSPA, we could not detect the function.

On the other hand, we believe that the PGE2 dynamics and its regulation mechanism in the normal epidermis would be worth reporting to researchers in the field.

1. The authors do not explain further why in certain cells of the cell clusters Ca2+ signals occur spontaneously and thus trigger the phenomenon. What triggers these Ca2+ changes? And why could this be linked to certain cell functions and functional changes?

At this moment, we do not have a clear answer or model for the comment although the calcium transients have been reported in the epidermis (https://doi.org/10.1038/s41598-018-24899-7). Further studies are needed and we will pursue this issue as a next project.

1. What explains the radius and the time span of the radial signal continuation? To what extent are these factors also related to the degradation of PGE2? The work could be stronger if such questions and their answers would be experimentally integrated and discussed.

We agree with the reviewer’s comment. Although we have intensively studied that point, we have omitted the results because of its complications. In HeLa cells, but not MDCK cells, we demonstrate the meaning of the radius of RSPA (https://pubmed.ncbi.nlm.nih.gov/37813623/)

1. The authors could consider whether they could investigate the subcellular translocation of cPLA2 in correlation with cytosolic Ca2+ signals using GFP technology and high-resolution fluorescence microscopy with their cell model.

Actually, we tried to monitor the cPLA2 translocation using GFP-tagged cPLA2. However, the translocation of GFP-cPLA2 was detected, only when the cells were stimulated by calcium ionophore. At this point, we have concluded that the quantitative analysis of cPLA2 translocation would be difficult.

**Reviewer #3 (Recommendations For The Authors):**
1. "The cell density in the basal layer is approximately 2x106 cells cm-2, which is markedly higher than that in MDCK cells (Fig. 2D). It is not clear whether this may be related to the lower frequency (~300 cm-2 h-1) and smaller radius of RSPA in the basal layer cells compared to MDCK cells (Fig. 2E)." Wasn't the relationship with cell density the opposite, higher density higher frequency? Isn't then this result contradicting the "cell density rule" that the authors argue is there in the in vitro system? The authors need to revise their interpretation of the data obtained.

We agree with the reviewer’s comment. Currently, we do not find the "cell density rule" in mouse epidermis. It would be difficult to identify common rules between mouse epidermis and MDCK cells. However, although it is descriptive, we believe it is worth comparing the MDCK results at this moment.

1. Similarly, the authors over conclude on the explanation of lack of change in the size of RSPA size when the change in fluorescence for the calcium reporter surpasses a threshold by saying that "This observation indicates that the amount of PGE2 secretion is predetermined and that there is a threshold of the cytoplasmic calcium concentration for the triggered PGE2 secretion." First, the study does not really measure directly PGE2 secretion. Hence, there is no way that they can argue that the level of PGE2 secreted is "predetermined". Instead, there could be an inhibitory mechanism that is triggered to limit further activation of PGE2 signaling/PKA in neighboring cells.

We agree with the reviewer’s comment. We have omitted the context.

1. To rule out a transcription-dependent mechanism in the apparent cell density-regulated sensitivity to PGE2, the authors need to inhibit transcription.We agree that our RNA-seq analysis would not 100% rule out the transcription-dependent mechanism. However, we believe that shutting down all transcription will show a severe off-target effect that indirectly affects the calcium transients and the PGE2-synthetase pathway. Therefore, our conclusion is limited.

1. EGF is reported to increase the frequency of RSPA but the change shown in Fig. 6F is not statistically significant, hence, EGF does not increase RSPA frequency in their experiments.

We have toned down the claim that EGF treatment increases the frequency (line172).

"Accordingly, the addition of EGF faintly increased the frequency of RSPA in our experiments, while the MEK and EGFR inhibitors almost completely abrogated RSPA (Fig. 6F), representing that ERK activation or basal ERK activity is essential for RSPA.“

1. The Discussion section is at times redundant with the results section. References to figures should be kept in the Results section.

We would like to argue in opposition to this comment. For readers, we believe that the reference to figures would be helpful and kind. However, if eLife recommends removing the reference from the Discussion section, we will follow the publication policy.

1. "Notably, the propagation of PKA activation, ~100 μm/min (Fig. 1H), is markedly faster than that of ERK activation, 2-4 μm/min (Hiratsuka et al., 2015)." The 2 kinase reporters are based on different molecular designs. Thus, it does not seem appropriate to compare the kinetics of both reporters as a proxy of the comparison of the kinetics of propagation of both kinases.

We think that we should discuss the comparison of the activity propagation between ERK and PKA. First, among many protein kinases, only ERK and PKA activities have been shown to spread in the epithelial cells. Second, both pathways are considered to be intercellular communication. Finally, crosstalk between these two pathways has been reported in several cells and organs.

1. In Figure 1E it is unclear what is significantly different from what. Statistical analysis should be added and reporting of the results should reflect the results from that analysis.1. In Figure 3F and G the color coding is confusing. In F pink is radius and black is GCaMP6 and in G is RSPA+ and - cells. The authors should change the color to avoid ambiguity in the code.

We have amended the panels.

1. In Fig. 5C, how do they normalize per cell density if they are measuring radius of the response?

In Fig5C, we just measure the increment of FRET ratio in the view fields.

1. In Fig. 5D, what is the point of having a label for PTGER3 if data were not determined (ND)?

We have added what N.D. means.

“N.D. represents Not Detected.”

1. It is important to assess whether ERK activation depends of PGE2 signaling to better place ERK in the proposed signaling pathway. In fact, the authors argue that "ERK had a direct effect on the production of PGE2." But it could be that ERK is downstream PGE2 signaling instead.

It could be possible in other experimental conditions via EP1 and/or EP3 pathways. However, we never detected an effect of RSPA on ERK activity by analyzing our imaging system. In addition, treatment with NSAIDs or COX-2 depletion, which completely abolishes RSPA, did not affect ERK wave propagation. Thus, in our context, we concluded that ERK is not downstream of PGE2. This notion is also supported by the NGS results in Fig. 5D.

We have refrained from discussing the pathway of PGE2-dependent ERK activation because it would be redundant.

1. The authors need to explain better what they mean by "AND gate" if they want to reach a broad readership like that of eLife

We have modified the legend to explain the “AND gate” as much as possible (line639).

“Figure 7: Models for PGE2 secretion.

The frequency of calcium transients is cell density-dependent manner. While the ERK activation wave is there in both conditions. Because both calcium transient and ERK activation are required for RSPA, the probability for PGE2 secretion is regulated as “AND gate”. ”

1. In Fig. 5D, "The average intensity of the whole view field of mKate2 or mKOκ, at 20 to 30 min after the addition of PGE2, was applied to calculate the mKate2/mKOκ ratio." But this means that overlapping/densely plated cells in high density will show stronger changes in fluorescence. This should be done per cell not per field of view. It is obvious that the higher density will have more dense/brighter signal in a given field of view.

We are sorry for the confusion. The cell density does not affect the FRET ratio, although the brightness could be changed. A typical example is Fig1D. Thus, we are sure that our procedures represent the PKA activity in plated cells.

1. In Fig. 6B the authors need to explain how were the "randomly set positions" determined.

We have modified the legend section as below (line618).

“The ERK activities within 10 µm from the center of RSPA and within 10 µm from randomly set positions with a random number table generated by Python are plotted in the left panel. Each colored dot represents an average value of an independent experiment.”

1. Sentences 314-318 are repeated in 318-322.

We deeply appreciate the reviewer’s comment and have amended